# Remifentanil Alleviates Propofol-Induced Burst Suppression without Affecting Bispectral Index in Female Patients: A Randomized Controlled Trial

**DOI:** 10.3390/jcm8081186

**Published:** 2019-08-08

**Authors:** Dahye Jung, Sungwon Yang, Min Soo Lee, Yoonki Lee

**Affiliations:** 1Department of Anesthesiology and Pain Medicine, Changwon Fatima Hospital, Changwon 51394, Korea; 2Department of Anesthesiology, Seoul St. Mary’s Hospital, College of Medicine, The Catholic University of Korea, Banpo-daero, Seocho-gu, Seoul 06591, Korea

**Keywords:** bispectral index, burst suppression, propofol, remifentanil

## Abstract

The bispectral index is affected by various factors, such as noxious stimuli and other drugs, such as muscle relaxants. The burst suppression ratio from bispectral index monitoring is correlated with electroencephalographic burst suppression, which is associated with deep anesthesia, metabolic disorders, and brain injury. We assessed patients undergoing total intravenous anesthesia and examined the effects of remifentanil on the bispectral index, burst suppression ratio, and hemodynamic changes immediately after loss of consciousness with propofol. Seventy American Society of Anesthesiologists physical status class I and II Korean female patients scheduled for general anesthesia were administered propofol with an effect-site concentration of 5 μg/mL, using a target-controlled infusion (TCI). After losing consciousness, patients received either saline or remifentanil at an effect-site concentration of 5 ng/mL for 10 min. During this period, we recorded the bispectral index values, including burst suppression ratio, blood pressure, and heart rate. With remifentanil infusion, burst suppression ratios were lower (*p* < 0.01) but bispectral values were not different. The burst suppression ratio was significantly different at 6, 7, 8, and 10 min after remifentanil infusion (*p* < 0.05). In female patients with propofol-induced unconsciousness, remifentanil alleviated the burst suppression ratio without affecting the bispectral value.

## 1. Introduction

The bispectral index (BIS) analyzes electroencephalographic waves and represents a numeric value to objectively evaluate the level of consciousness. It is used widely during sedation and general anesthesia [1]. These values, however, are affected by various factors during anesthesia and surgery, such as tracheal intubation, surgical stress [2,3,4], or muscle relaxants [5,6].

Total intravenous anesthesia, usually consisting of propofol and remifentanil without a muscle relaxant, is the current gold standard during intraoperative neurophysiological monitoring, especially for brain or spinal surgeries [7]. In these circumstances, the BIS can be used to monitor anesthetic depth and adjust the drug dosage.

Remifentanil has been found to decrease the BIS value during tracheal intubation and skin incision, even in the absence of other stimuli [2,8,9]. However, previous reports have found that remifentanil does not affect BIS values [10,11]. Muscle relaxants have complex effects that can also affect BIS values [5,6]. Most studies dealing with general anesthesia have examined the effects of remifentanil on the BIS in the presence of muscle relaxants, mostly after tracheal intubation and surgical incision [2,8,9,10], although some assessed propofol-sedated patients without muscle relaxants [12].

The burst suppression describes an electroencephalographic pattern consisting of a continuous alteration between high-voltage slow-wave and depressed electrographic activity [13]. Also, the burst suppression ratio, derived from BIS monitoring, is correlated with the burst suppression and could be an indicator of anesthesia that is too deep [14]. Several reports have suggested that the burst suppression ratio is associated with intraoperative metabolic disorders or brain injury [15,16,17], and postoperative prognosis and delirium [18,19,20].

Propofol and remifentanil both lower blood pressure and heart rate, and concomitant administration further reduces blood pressure, affecting pulse rate [2,8].

The primary objective of the current study was to examine the effects of remifentanil on the BIS in the absence of muscle relaxants and noxious stimuli in patients with propofol-induced unconsciousness. Next, we observed the change in the burst suppression ratio with or without remifentanil infusion. Finally, we examined the effect of remifentanil on the mean blood pressure and heart rate

## 2. Method

This study was registered on 15 October 2013 at http://cris.nih.go.kr (KCT0000880) and was approved by the Institutional Review Board of Changwon Fatima Hospital (IRB No.14-03). After obtaining written permission, 70 female patients over 20 and below 55 years of age of American Society of Anesthesiology (ASA) physical status class I or II who were scheduled for total intravenous general anesthesia in gynecological surgery were included in this prospective, placebo-controlled study. This study was conducted between 2014 and 2015. The exclusion criteria were patients with cardiovascular disease, diabetes mellitus, or neurologic disease; those who were taking medications that affect the central nervous system; anticipated difficulty in airway management; body mass index over 30; or refusal to participate in the study.

### 2.1. Study Design and Data Collection

Premedication was omitted, and standard monitoring, including electrocardiography, non-invasive blood pressure, and pulsoximeter, was initiated on arrival to the operating room. A BIS Quatro sensor was attached to the patient’s forehead according to the manufacturer’s guidelines and connected to a BIS-Vista monitor (Aspect Medical Systems, Inc., Natick, MA, USA). Patients were randomized into either the remifentanil or control group based on computer-generated random numbers with a 1:1 allocation ratio (*n* = 35 for each group), by a person who was not involved with the recruitment and selection of patients.

After administration of glycopyrrolate 0.2 mg and 1% lidocaine 20 mg, propofol was infused to the target effect-site concentration of 5 μg/mL using an infusion device (Orchestra^®^ Base Primea, Fresenius Vial, Brezins, France) with the Schnider pharmacokinetic model. When the target concentration reached 5 μg/mL, the patient’s loss of consciousness was assessed via verbal responses and eyelid reflexes. If the patient did not lose consciousness at the target concentration of 5 μg/mL within 5 min, the patient was excluded from the study.

After the patients lost consciousness, they were allowed to breathe spontaneously but were ventilated with a facial mask to maintain end-tidal ETCO2 30~39 mmHg. With continuous infusion of propofol at that concentration, the remifentanil group patients were administered remifentanil at a target effect-site concentration of 5 ng/mL using the Minto model, and the control group received a saline infusion instead of remifentanil. The patient and the person administering the drug were blind to the type of group and the drug administered.

BIS values, including burst suppression ratio, mean blood pressure, and heart rate, were recorded every 1 min for 10 min immediately after the remifentanil infusion at a target effect-site concentration of 5 ng/mL. If the patient’s systolic blood pressure decreased below 80 mmHg, fluid loading and the reverse Trendelenburg position were initially used; ephedrine was used when the blood pressure was not restored to above 80 mmHg. All data were recorded by anesthetic nurses blind to group allocation. After completion of the study, a muscle relaxant was administered, and the patient was intubated and managed at the discretion of the attending anesthesiologist.

### 2.2. Statistical Analysis

Sample size was calculated based on a pilot study (*n* = 12) of BIS values, the primary outcome, with mean difference of 8.26, and standard deviation (SD) of 15.23, respectively. Considering alpha = 0.05, power = 80%, and drop-out rate for 10%, the estimated sample size was 70.

Continuous variables were expressed as mean ± SD, and categorical variables were expressed as numbers, including percentages. To assess the effects of remifentanil, repeated measure ANOVA tests were used, including post-hoc analysis (Friedman test). To compare the variables at each time point, an independent *t* test, Mann–Whitney U test, or Fisher’s exact test was used, if appropriate. All statistical analyses were performed by R version 3.4.4 (https://www.r-project.org/) and *p*-values below 0.05 were considered statistically significant.

## 3. Results

Three patients in the control group and two patients in the remifentanil group did not lose consciousness at the propofol effect-site concentration of 5 μg/mL during 5 min of observation. Therefore, a total of 65 patients participated in the study (Figure 1). The two groups did not have any significant differences in demographics or preoperative variables (Table 1). All patients had a burst suppression ratio of zero when the BIS was attached.

BIS values decreased over time for both the control and remifentanil groups compared with the preoperative value (*p* < 0.05), but there was no difference between the two groups. Also, the BIS values did not show any difference at any time point between the two groups during the study period (Figure 2). The burst suppression ratios were significantly lower in the remifentanil group compared with the control group (*p* < 0.01). In addition, the ratios in the remifentanil group were significantly lower than in the control group at 6, 7, 8, and 10 min (*p* < 0.05) (Figure 3). The number of patients showing burst suppression (burst suppression ratio(BSR) ≥ 1) was significantly more in the control group compared with the remifentanil group (*p* < 0.01). The numbers of patients with burst suppression was significantly different at 7, 8, and 10 min (*p* < 0.05) (Table 2). The median values and interquartile ranges of the burst suppression ratios in patients showing BSR ≥ 1 at the end of experiments were 5 and 7 in the control group, and 2 and 0 in the remifentanil group, respectively.

Seven patients in the remifentanil group required ephedrine to maintain their blood pressure, compared with zero patients in the control group. Mean arterial blood pressure decreased over time (*p* < 0.05), but there was no difference between the groups at any time point (Figure 4). Heart rate decreased over time, but no significant difference was observed between them. However, at 7, 8, 9, and 10 min, the remifentanil group had a decreased heart rate compared with the control group (*p* < 0.05) (Figure 5).

## 4. Discussion

In this study, we found no effect of remifentanil on the BIS in patients with propofol-induced unconsciousness, even in the absence of noxious stimuli and muscle relaxants. Also, the mean blood pressure and heart rate were not affected, although ephedrine was more likely to be required following remifentanil infusion. By contrast, burst suppression ratios were lower following remifentanil infusion and more prominent at the later time points.

Considering the gender difference in opioid sensitivity [21,22], we included only female patients in this study. In a previous study, we found that the EC50 and EC95 of the propofol effect-site concentration for the loss of consciousness in Korean patients was 4.11 (95% confidence interval (CI): 3.61–4.50) and 4.57 (95% CI: 4.31–8.38), respectively [23]. Based on these results, we chose a propofol effect-site concentration of 5 μg/mL, which caused about 93% of patients to lose consciousness. In Chinese patients, the effect-site of remifentanil concentration to achieve a 95% probability of nonresponse to tetanic stimulus, which was usually used in lieu of skin incision (predicted effect-site concentration C95), was shown to be 5.1 ng/mL [24]. Therefore, a remifentanil effect-site concentration of 5 ng/mL was used in this study.

At light anesthetic depths, like sedation, controversies exist, whether remifentanil decreased BIS [4,12] or not [11]. However, during deeper anesthesia, such as general anesthesia, remifentanil is known not to affect the BIS values [2], although one report stated that a 2 μg/mL remifentanil bolus under total intravenous anesthesia with propofol decreases the BIS [9]. Interestingly, the studies in deeper anesthesia were conducted after intubation with the aid of muscle relaxants, which may have affected the BIS values [5,6,25]. In addition, the propofol and remifentanil doses used in the present study were somewhat higher than that for sedation, and the remifentanil was infused, rather than bolus, which might cause more constant concentration. Therefore, these factors could have contributed to having no significant effect of remifentanil on BIS.

Although this study was conducted with unstimulated, unparalyzed, and unconscious patients, the common patients in the operating room usually receive lots of stimuli, such as intubation or surgical stress. Usually, catecholamine is released more and in turn, this increases the BIS values [3]. Then, like remifentanil, opioids are given to suppress somatic stress and adrenergic response to stimulation. In this situation, the interactions between propofol and remifentanil on BIS could be observed [4]. Thus, it is possible to postulate that the effects of propofol used were more pronounced in unstimulated patients, in whom the remifentanil’s effect on BIS was overwhelmed, than in stimulated patients. However, the interactions mentioned need further verification.

The observation of intraoperative burst suppression can predict postoperative delirium [19] and Soehle et al. reported that the intraoperative burst suppression ratio may help to identify patients who are at risk for postoperative delirium following cardiac surgery [20].

Although we found a lower suppression ratio when remifentanil was infused, no difference could be found in the BIS. However, when considering a suppression ratio of 40% or more is invariably and linearly correlated [26,27], it is possible to postulate that the relatively low value of suppression might not be able to affect the bispectral values. It remains unclear how the BIS values are computed for burst suppression ratios <40%.

In cardiac surgical patients, Soehle et al. [20] found that an intraoperative higher burst suppression ratio was associated with postoperative cognitive dysfunction. According to their study, they reported BSR values of 1.24% in patients who developed delirium postoperatively compared with those values of 0.44% in patients without delirium. Furthermore, the presence of burst suppression itself was associated with an increased mortality in critically ill patients [28]. Therefore, low values or even the presence of BSR values may forecast an early warning of the following neurologic outcome, especially in major surgical patients and critically ill patients.

In this study, the burst suppression ratio of the propofol-infused patients was lower when remifentanil was infused simultaneously. The changes remifentanil produces in the electroencephalography(EEG) are typical of μ-receptor agonists. Also, the EEG shifted from low-amplitude, high-frequency activity during baseline to high-amplitude, low-frequency activity during opioid infusion, culminating eventually in delta activity at maximal drug effect [29]. However, the effects of remifentanil on EEG in propofol anesthesia were different, depending on the depth of anesthesia and dose of remifentanil used. During deep anesthesia, remifentanil showed dose-dependent increased activity in the extended alpha band (7–14 Hz) and decreased activity in the delta band (0.5–4 Hz) [29], partly explaining the lower suppression ratio when remifentanil was infused. So, we cautiously speculate that the maintenance of a propofol concentration of 5 μg/mL induced a deep anesthetic state, resulting in an electroencephalic change, which might affect the appearance of burst suppression. It is necessary to examine the relationship between remifentanil effects on the propofol-induced burst suppression ratio and clinical outcome factors, such as postoperative delirium and other sequelae.

In this study, seven patients required ephedrine to maintain blood pressure following remifentanil infusion. Therefore, we further analyzed whether ephedrine affected the BIS and burst suppression ratio values by excluding the patients who were administered ephedrine. The results did not differ, suggesting the ephedrine might not affect the BIS and burst suppression.

Opioids typically decrease heart rate and mean arterial pressure. Propofol is also known to lower blood pressure and heart rate. In this study, there was no significant difference in mean arterial blood pressure, whether remifentanil was infused or not. However, seven patients required ephedrine to maintain blood pressure following remifentanil infusion, indicating that ephedrine may affect these results. We observed a further decrease in heart rate later, when remifentanil was administered.

The current study was not without limitations. First, we used fixed doses of propofol and remifentanil, so varying the dosage may alter the results. Second, we infused remifentanil for only 10 min, so the effects of a longer period of remifentanil infusion are unknown. Additionally, the low values of the burst suppression ratio preclude any definitive conclusion about subtle BIS value modification. Finally, the results cannot be generalized to male patients and other types of surgery.

## 5. Conclusions

In conclusion, remifentanil infusion at a 5 ng/mL effect-site concentration does not affect the BIS value in the absence of muscle relaxants and noxious stimuli in unconscious female patients with a propofol effect-site concentration of 5 μg/mL. Suppression ratios were lower when remifentanil was infused. Although the mean blood pressure and heart rate were not affected, heart rate decreased more following the infusion of remifentanil later in this study.

## Figures and Tables

**Figure 1 jcm-08-01186-f001:**
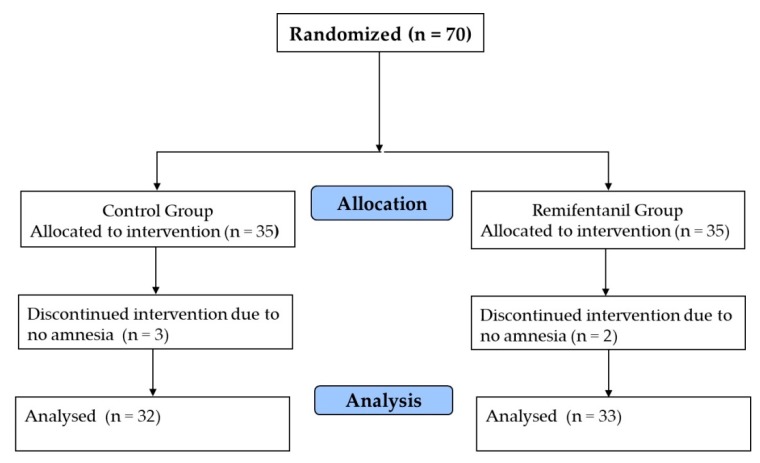
Randomization and study flow diagram.

**Figure 2 jcm-08-01186-f002:**
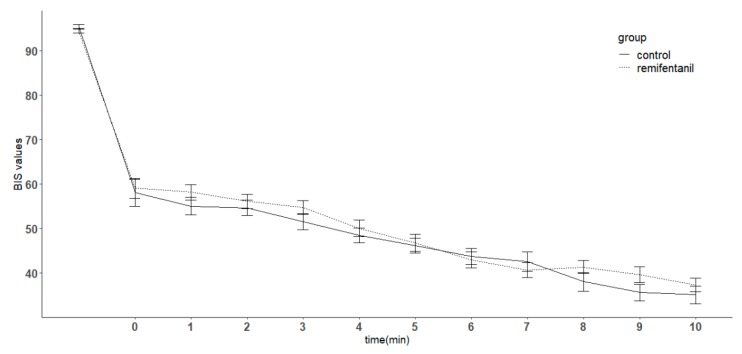
Changes in the bispectral index (BIS) in patients with or without remifentanil infusion. In unconscious patients with propofol at the target effect-site concentration of 5 μg/mL, a remifentanil target effect-site concentration of 5 ng/mL was infused in the remifentanil group throughout the study period. The control group received normal saline instead of remifentanil. BIS decreased over time for both the control and remifentanil group compared to the preoperative value (*p* < 0.05), but there was no difference between the two. Error bar indicates standard error. In the *x* axis, time point 0 indicates the time the remifentanil infusion was started at an effect-site concentration of 5 ng/mL.

**Figure 3 jcm-08-01186-f003:**
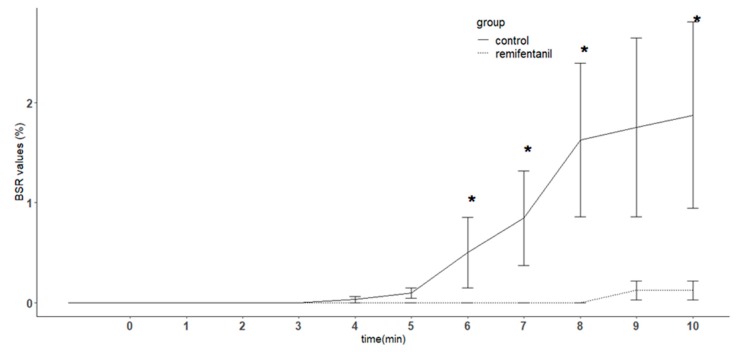
Changes in the burst suppression ratio (BSR) in patients with or without remifentanil infusion. In unconscious patients with propofol at the target effect-site concentration of 5 μg/mL, a remifentanil target effect-site concentration of 5 ng/mL was infused in the remifentanil group throughout the study period. The control group received normal saline instead of remifentanil. The BSR was less in the remifentanil group compared to the control group (*p* < 0.01), and the BSR in the remifentanil group was significantly lower than the control group at 6, 7, 8, 10 min (*p* < 0.05). Error bar indicates standard error. In the *x* axis, time point 0 indicates the time the remifentanil infusion was started at an effect-site concentration of 5 ng/mL. * denotes *p* < 0.05 compared with control group in corresponding time point.

**Figure 4 jcm-08-01186-f004:**
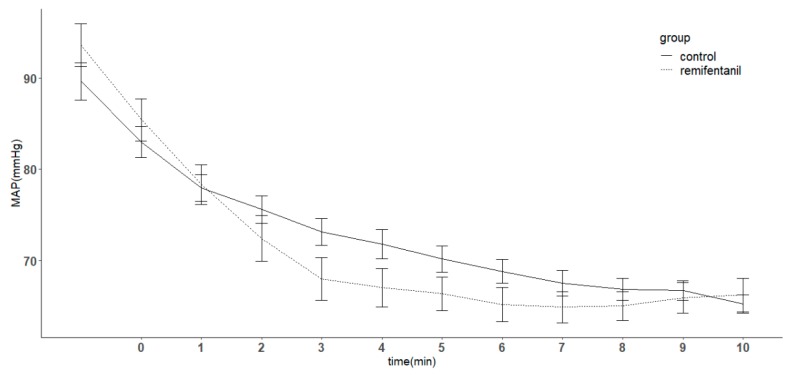
Changes in mean arterial pressure (MAP) in patients with or without remifentanil infusion. In unconscious patients with propofol at the target effect-site concentration of 5 μg/mL, a remifentanil target effect-site concentration of 5 ng/mL was infused in the remifentanil group throughout the study period. The control group received normal saline instead of remifentanil. MAP decreased over time (*p* < 0.05). There was no difference between the remifentanil group and control group. During the study, the two groups showed no difference at any time point. Error bar indicates standard error. In the *x* axis, time point 0 indicates the time the remifentanil infusion was started at an effect-site concentration of 5 ng/mL.

**Figure 5 jcm-08-01186-f005:**
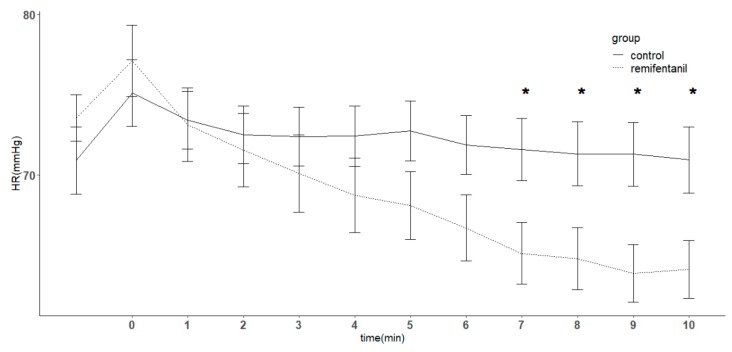
Changes in heart rate (HR) in patients with or without remifentanil infusion. In unconscious patients with propofol at the target effect-site concentration of 5 μg/mL, a remifentanil target effect-site concentration of 5 ng/mL was infused in the remifentanil group throughout the study period. The control group received normal saline instead of remifentanil. HR decreased over time and an interaction was observed (*p* < 0.05). There was no difference between the remifentanil group and control group, although the remifentanil group showed a significantly lower heart rate at 7, 8, 9, and 10 min. Error bar indicates standard error. In the *x* axis, time point 0 indicates the time the remifentanil infusion was started at an effect-site concentration of 5 ng/mL. * denotes *p* < 0.05 compared with the control group at the corresponding time point.

**Table 1 jcm-08-01186-t001:** Demographics and Baseline Hemodynamics of the Study Patients.

	Control (*n* = 32)	Remifentanil (*n* = 33)	*p*-Value
Age (year)	43.6 (7.7)	41.5 (7.8)	0.28
Height (cm)	160.1 (5.3)	160.3(5.3)	0.84
Weight (kg)	56.8 (6.6)	57.9(8.5)	0.54
ASA physical status			
I	24 (75.0)	24 (72.7)	1
II	8 (25.0)	9 (27.3)	-
BIS value	95.4 (2.5)	94.5 (2.8)	0.13
Mean blood pressure	89.6 (11.4)	93.6 (13.4)	0.20
Heart rate	70.9 (11.9)	73.6 (8.3)	0.30

Values are expressed as mean (SD) or numbers (%). ASA indicates American Society of Anesthesiologists; BIS indicates bispectral index.

**Table 2 jcm-08-01186-t002:** Number of patients showing BSR ≥ 1% during 10 min of propofol-induced unconsciousness in the control and remifentanil groups.

	0~3 min	4 min	5 min	6 min	7 min	8 min	9 min	10 min
**Control (*n* = 32** **)**	0	1	3	4	5	7	7	8
**Remifentanil (*n* = 33)**	0	0	0	0	0	0	2	2
***p*-value**	1	0.492	0.113	0.053	<0.05	<0.05	0.082	<0.05

Control and remifentanil group showed significant difference (*p* < 0.01). *p*-values indicate the difference at each time-point.

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
