# Peer review of "Remifentanil Alleviates Propofol-Induced Burst Suppression without Affecting Bispectral Index in Female Patients: A Randomized Controlled Trial"

_jcm, 2019, doi:10.3390/jcm8081186_

Round 1
Reviewer 1 Report
Dear authors,
Thanks for submitting your work to the journal. You describe an investigation in seventy ASA 1 and 2 patients receiving propofol and remifentanil or saline, and monitored by the bispectral index (BIS) and the burst suppression ratio (BSR). You conclude that the remifentanil alleviates the BSR without affecting the BIS.
The work is well presented, may have been improved at the design stage by the use of a complete EEG monitoring, but this does not mean that this is not interesting. Even more, the results are intriguing as one may expect the opposite, i.e. lower BIS and/or higher BSR with remifentanil.
Of course, the low values of the BSR precludes any definitive conclusion about subtle BIS values modifications. This is acknowledged by the authors but may be better mentioned in the limitations section.
An important paragraph, even if speculative, may be added in the discussion regarding the reason why remifentanil may modify the EEG pattern. The authors speculate on the protective effect of this, but one may also defend a deleterious effect of an opioids induced excitoxicity, e.g. through the activation of the NMDA receptors.
The manuscript may benefit from a rereading for persisting typos.
Reviewer 2 Report
General comments
Basically, this is an small series of RCT. Authors reported remifentanil can alleviate propofol-induced burst suppression without affecting bispectral Index. They concluded remifentanil infusion at a 5 ng/mL effect-site concentration does not affect the BIS value in the absence of muscle relaxants and noxious stimuli. The overall design of this RCTs is good, the methods are adequately described by the authors and the conclusions are justified by the data. However, the manuscript suffered from some problems need to be clarify.
I have some comments to challenge the authors to make this work even more attractive (I hope?)
1. BSR is a time domain analysis in BIS, while Beta ratio and Bi-spectrum analysis are another two frequency domain analysis of BIS value, btw, can you tell me why there are difference among BSR with no differences in BIS? Is the sample size too small?
2. Only female patient in this RCT, please add it to the title.
3. Page 6 Line 174 “To exclude gender-related bias, we included only women in this study.”, only women since not a good method to exclude gender-related bias, how about man?
4. Page 6 Line 178-181, why you used the remifentanil effect-site concentration of 5 ng/mL which is relative to a tetanic stimulus in this no noxious stimuli study?
5. Page 6 Line 199-222, review the BSR seems a little bit redundant and should be shrinked
6. Why the observation time is set to 10 minutes since there are still a decrease trend after 10 minutes in Figure 2?
7. Why you do not use figure instead table to show the results of BSR just like the BIS,MAP, and Heart rate?
Round 2
Reviewer 2 Report.
Author Response
Thank you for reviewing.